# Gender Differences in Mental Rotational Training Based on Computer Adaptive Tests

**DOI:** 10.3390/bs13090719

**Published:** 2023-08-29

**Authors:** Hanlin Wang, Linghe Li, Pan Zhang

**Affiliations:** Department of Psychology, Hebei Normal University, Shijiazhuang 050024, China; wanghanlin@hebtu.edu.cn (H.W.); annieturtleli@163.com (L.L.)

**Keywords:** mental rotation, three-down/one-up staircase method, gender difference

## Abstract

Mental rotation tasks have been widely used to assess individuals’ spatial cognition and the ability to mentally manipulate objects. This study employed a computerized adaptive training method to investigate the behavioral performance of participants of different genders in mental rotation tasks with different rotation angles before and after training. A total of 44 Chinese university students participated in the experiment, with the experimental group undergoing a five-day mental rotation training program. During the training phase, a three-down/one-up staircase procedure was used to adjust the stimulus levels (response time) based on participants’ responses. The results showed that the training had a facilitative effect on the mental rotation ability of both male and female participants, and it was able to eliminate the gender differences in mental rotation performance. Regarding the angles, we observed that the improvement in the angles involved in the training was significantly higher compared to untrained angles. However, no significant differences in improvement were found among the three trained angles. In summary, these findings demonstrate the effectiveness of computerized adaptive training methods in improving mental rotation ability and highlight the influence of gender and angles on learning outcomes.

## 1. Introduction

Mental rotation ability, as a cognitive capacity for visual–spatial representation manipulation, pertains to an individual’s aptitude in mentally envisioning the rotation of objects across different angles. It constitutes a fundamental component of spatial ability and encompasses three primary facets: perception, rotation, and decision-making [1,2,3]. Contemporary cognitive science research on mental rotation continuously provides insights into the underlying processes of spatial perception and representational manipulation [4], revealing noteworthy associations between mental rotation and proficiencies in diverse domains. Notably, mental rotation ability has demonstrated its potential as a precise predictor of athletic performance [5]. Moreover, students’ visuospatial abilities, encompassing mental rotation, have been found to facilitate cognitive operations in spatial contexts, thereby contributing to academic achievements in subjects such as mathematics and chemistry [6,7,8]. For example, Wai et al. [9] conducted a longitudinal study on a cohort of 400,000 participants from U.S. high schools, tracking them for 11 years to examine the significance of spatial ability in educational pursuits and STEM domains. The research findings supported that spatial ability assessed during adolescence is a critical predictor of obtaining higher educational credentials and pursuing STEM careers. Recent empirical inquiries have also emphasized the crucial role of mental rotation capabilities in the context of human-controlled rendezvous and docking missions during human spaceflight endeavors [10]. This highlights the practical applications of studying mental rotation for comprehending spatial cognitive abilities, with a particular emphasis on representational skills. Thus, the study of mental rotation in normal groups not only holds significant theoretical implications but also has pragmatic value in understanding and enhancing spatial cognitive abilities across diverse domains, including academic performance and specialized tasks in the STEM and spaceflight fields.

The ability to rotate objects and scenes is critical to many professional skills, so how to improve mental rotation performance has become a significant focus of researchers. It is well established that performance on standard mental rotation tasks improves with training [11]. The study conducted by Lizarraga et al. [12] has also revealed that participants’ mental rotation abilities were enhanced, leading to improved performance on an untrained visualization task, as a result of mental rotation training and feedback. Nevertheless, the conventional training paradigms still have areas for further improvement [10]. First, most of the training for mental rotation uses the method of constant stimuli (MCS), in which both correctness and response time are dependent variables. It leads to a trade-off between correct rate and reaction time when analyzing the data, and it is not easy to visually compare the differences before and after training [13,14]. Second, the MCS, in which all subjects use the same stimulus intensity, may make the training task too hard for some subjects but too easy for others [15]. In contrast, according to educational psychology theory, learning is most likely to occur when task difficulty is moderate [16,17,18]. In the current study, we chose the three-down/one-up staircase method, which estimates the threshold at 79.3% correct by adjusting the stimulus level (e.g., duration, luminance, or contrast) based on the subject’s responses [19]. Specifically, this method starts with an initial stimulus level typically determined based on the subject’s baseline performance. Then, depending on the subject’s responses in each trial, the stimulus level is increased or decreased using a fixed or adaptive step size. The procedure continues until a predetermined number of trials or reversals is reached, at which point it stops. This training method is commonly employed in perceptual learning studies, where stimulus difficulty is dynamically adjusted based on individual responses, effectively addressing the limitations of fixed difficulty levels observed in previous methods [20,21]. In perceptual learning, this method facilitates improvement by adjusting the level of stimulus attributes and providing feedback, resulting in a certain level of accuracy throughout training. Considering that mental rotation training similarly requires achieving more personalized and efficient training effects while aiming to mitigate the impact of speed–accuracy trade-off, adopting this method can reasonably predict positive outcomes in mental rotation training. The adaptability of this method ensures a personalized and efficient training experience, maximizing participants’ learning and performance.

Additionally, mental rotation produces more significant sex differences compared to other factors of spatial ability. Some extensive meta-analyses and large-sample studies have found significant and consistent sex difference in mental rotation, which indicates that males perform better than females on mental rotation tasks [22,23,24,25]. However, it has also been suggested that the magnitude of this sex difference varies depending on environmental factors [26]. Different results have also been obtained regarding gender differences in mental rotation visuospatial training. Some studies have concluded that males and females make consistent progress in mental rotation ability. Other studies suggest that learning improvement may be more pronounced in females after training, causing gender differences in the ability to disappear [27,28]. Some models attribute gender differences in mental rotation tasks to gender differences in the speed–accuracy trade-off. This explanation is based on findings that indicate gender differences in favor of male subjects decrease in effect size once time limits are removed from the test [29]. Some studies have explained that removing the time limit may have allowed females to utilize effective mental rotation strategies [30]. It has also been suggested in previous research that the reduction in gender differences observed in the training condition may be attributed to a ceiling effect in the male population. In this scenario, male subjects were unable to further improve their already good test performance after the removal of the time limit [31]. In summary, there are significant gender differences in mental rotation tasks, and these differences may also extend to the training outcomes of mental rotation. This study aims to further investigate the gender differences in mental rotation ability, their manifestation in training outcomes, and the underlying causes of these phenomena.

The classic study conducted by Shepard and Metzler [32] demonstrates a linear relationship between the time required for participants to perform mental rotation judgments and the angular difference between two objects. However, subsequent research has revealed that for smaller angles, the response time remains relatively stable, while for larger angles, there is a significant variation in response time. The response time curves exhibit a gentle onset followed by a steeper incline, suggesting a closer resemblance to quadratic or exponential function models [33]. Thus, it is likely that different strategies are employed during mental rotation tasks at different angles. Recent investigations by Martina Rahe et al. [34] suggest that individuals of different genders employ distinct strategies when performing mental rotation tasks. Consequently, it is worth exploring whether gender differences manifest in training outcomes at various angles. In this study, we further investigate the role of rotation angles in mental rotation training.

Although mental rotation ability has been extensively studied, there is still a need for a comprehensive investigation into the training effect on this cognitive capacity. Firstly, there is room for improvement in traditional mental rotation training, including issues related to the trade-off between speed and accuracy and the limitation of difficulty not matching the participants’ levels. Introducing adaptive training methods is expected to overcome these limitations and provide a more personalized and effective training experience for each participant. Secondly, gender differences in mental rotation ability have long been an intriguing topic. However, the impact of training on gender differences in mental rotation ability has not been extensively explored. Understanding the interaction between gender and training outcomes can reveal potential interventions to mitigate gender-related cognitive disparities. Lastly, the role of rotation angles in mental rotation training has received limited attention. Investigating the influence of rotation angles on mental rotation training can provide deeper insights into the specific effects of training on different aspects of mental rotation ability. In this study, a computerized adaptive training paradigm was used to perform a mental rotation training intervention of 40 min per day for five days in young adults. In summary, the aim of this research is to address the following research questions:Are computerized adaptive practice methods applicable to the training of mental rotation ability? Can they effectively improve performance on mental rotation tasks? In this study, we hypothesize that the training method employed is both effective and feasible;Are women disadvantaged in terms of their mental rotation ability? Does training have an impact on this gender difference? In this study, our hypothesis is that prior to training, female subjects would exhibit weaker mental rotation abilities compared to males. However, after training, the gender difference in mental rotation ability would diminish;Can training produce varying effects on mental rotation tasks based on different rotation angles? In this study, we hypothesize that the effectiveness of mental rotation training will vary depending on the rotation angles, and these variations will be influenced by gender differences.

## 2. Materials and Methods

### 2.1. Participants

We recruited 48 college students, among whom 4 participants dropped out during the study and were excluded, resulting in a final sample of 44 participants, with an equal distribution of 22 males and 22 females. Participants were matched with regard to sex and age. Then, it was decided by pseudorandom who in a pair was assigned to the experimental group (*n* = 22; mean age: 21.5; range 18–25 years; 11 women 11 men) and who to the control group (*n* = 22; mean age: 21.8; range 18–26 years; 11 women 11 men). All participants were right-handed native speakers of Chinese, with normal hearing, normal or corrected-to-normal vision, and no history of psychiatric or neurological disorders. Written consent was obtained from all participants, who were compensated for their participation after the experiment. The experiment was approved by the University Ethics Committee (approval number: 2021LLSC029).

### 2.2. Design and Materials

The experiment consisted of a mental rotation ability test and a mental rotation training task. The test task was to be performed once before and after the training. The two testing tasks were identical and were used mainly to evaluate the effect of training on mental rotation ability. The training task took place over five days between two tests, with one session per day of approximately 40 min each. Participants in the control group only performed the test tasks before and after the training in order to eliminate the practice effects of the tasks in the pretest.

The experimental stimulus material for the mental rotation ability test consisted of 280 two-dimensional pictures composed of black and white blocks, presented in pairs to the participants simultaneously. The relative rotation angles were 0°, 60°, 120°, and 180°. It is important to note that the rotation of these two pictures occurred within the picture plane (2D), meaning the rotation only took place in the two-dimensional plane of the images themselves. Participants were instructed to use the “F” and “J” keys as response keys to complete the task (Figure 1a). The experimental program was written in E-prime 3.0 and displayed on a computer.

The experimental stimulus material in the mental rotation training task was the same as in the test task, and two pictures with different rotation angles were presented to the subjects simultaneously, with relative rotations of 60°, 120°, and 180°. The “←” and “→” keys on the keyboard were used as response keys (Figure 1b), and the subjects wore HUAWEI original semi-in-ear wired headphones throughout the training. A difficulty-adaptive computer test was used for the experimental procedure in the training phase. The MATLAB programs with PsychToolbox extensions presented the practical task and recorded behavioral data.

## 3. Procedure

### 3.1. Mental Rotation Ability Test Task

Each participant experimented individually. Participants sat in front of a computer screen and placed their index fingers on the “F” and “J” keys. They were asked to look at each pair of pictures, imagine rotating the picture on the right until it was visually identical to the left, and then determine whether the two pictures were identical or mirror images. Pressing the “F” key indicated that the two pictures were similar while pressing the “J” key indicated that the two pictures differed. Participants were asked to respond as quickly and accurately as possible.

The experiment consisted of a practice stage and a formal test stage. First, participants read the experiment description and then performed eight practice trials. The duration of each trial was 4500 ms. If the participant answered incorrectly or did not make a keystroke judgment during the presentation time, the feedback “wrong” was displayed. A 500 ms red “+” gaze point was presented before each trial. The purpose of the practice stage was to help participants become familiar with the experiment, and no data were collected.

The formal experiment was identical to the practice experiment, but participants did not receive feedback after making their responses. The formal experiment consisted of 20 blocks, each containing six trials, with six different pairs of pictures, each with the same relative rotation angle. After each block, there was a 15 s pause. In total, 120 pairs of pictures were presented. Participants’ correct rates and response times were recorded (Figure 2).

### 3.2. Mental Rotation Training Task

The participants sat in front of a computer screen and placed index fingers on the “←” and “→” keys. Participants were asked to look at each pair of pictures, imagine rotating the picture on the right until it aligned with the picture on the left, and then determine whether the two pictures were identical or mirrored. Each group of pictures appeared with a “ding” sound. If the answer was correct, a dot appeared in the center of the screen, and the headphones made a “beep” sound. If the answer was wrong, only a dot appeared in the center of the screen, but there was no sound.

The experiment consisted of six blocks, each with 60 training trials and a 60 s break between every two blocks. A total of 360 sets of images was presented, randomly grouping the three rotation angles (60°, 120°, 180°). Unlike the test task, there was a time progress bar above the pictures, which was automatically adjusted to display the remaining time for each individual trial based on the correctness of the subject’s answers (Figure 3). The initial setting time was 5 s for 60°, 7.5 s for 120°, and 10 s for 180°. It was determined based on the results of our preliminary experiment. A three-down/one-up staircase algorithm [19] was used to assess the thresholds (presentation time). This algorithm controlled the presentation time of the target image. This method decreased the presentation time by 10% (=the previous value×90%) if subjects correctly responded every three consecutive times and increased it by 10% (=the previous value×110%) if subjects made an incorrect response. Under this “three-down/one-up rule”, the staircase method could converge to 79.4% correct responses [19]. The trial defined a reversal or endpoint, after which the presentation time was increased or decreased. To ensure more reliable estimates [20,21], after excluding the first four or five inflection points to the threshold, we averaged the remaining even inflection points.

## 4. Results

### 4.1. Effects of Training on Mental Rotation Performance

To investigate the effectiveness of the training mentioned in this study, we employed a two (test) × two (group) design and conducted a repeated measures analysis of variance (ANOVA). The results of the variance analysis revealed a significant main effect of the test condition on accuracy (*F* (1, 42) = 49.14, *p* < 0.001, η_p_^2^ = 0.54). Post hoc analysis demonstrated that the accuracy after training (*M* = 0.92, *SD* = 0.09) was significantly higher than the accuracy before training (*M* = 0.83, *SD* = 0.10). Additionally, a significant interaction effect between the test condition and the group was observed (*F* (1, 42) = 5.84, *p* = 0.02, η_p_^2^ = 0.12). Subsequent simple effect analysis revealed no significant difference in accuracy between the control group and the experimental group before training (*p* > 0.05). However, after training, the accuracy of the control group was significantly lower than that of the experimental group (*F* (1, 42) = 7.81, *p* = 0.008, η_p_^2^ = 0.16).

Regarding reaction time, the variance analysis indicated a significant main effect of the test condition (*F* (1, 42) = 102.58, *p* < 0.001, η_p_^2^ = 0.71). Post hoc analysis revealed that the reaction time after training (*M* = 1814.38, *SD* = 336.49) was significantly faster compared to the reaction time before training (*M* = 2306.31, *SD* = 337.96). Furthermore, a significant interaction effect between the test condition and the group was observed (*F* (1, 42) = 13.01, *p* < 0.001, η_p_^2^ = 0.24). Subsequent simple effect analysis showed no significant difference in reaction time between the control group and the experimental group before training (*p* > 0.05). However, after training, there was a marginal trend indicating a higher reaction time in the control group compared to the experimental group (*F* (1, 42) = 3.70, *p* = 0.06, η_p_^2^ = 0.08) (see Figure 4).

### 4.2. Gender and Angle Differences in Mental Rotation Training Effects

These results indicate that training does have a positive effect on mental rotation ability. To advance our understanding of potential gender and angle differences in the training effect of mental rotation ability, we specifically selected the experimental group and employed a two (test) × two (gender) × four (angle) design to conduct a thorough analysis using repeated measurement analysis of variance (ANOVA). The results are shown in Table 1.

Considering reaction time as the dependent variable, we observed a significant interaction between the test and gender (*F* (1, 60) = 5.43, *p* = 0.03, η_p_^2^ = 0.21). Subsequent analysis revealed that during the pretest phase, the reaction time of female subjects was significantly longer than that of males (*F* (1, 20) = 4.58, *p* = 0.04, η_p_^2^ = 0.19), while in the post-test, the difference in response time of male and female subjects was no longer significant (*p* > 0.05) (see Figure 5). The analysis of accuracy as the dependent variable did not yield significant main effects of gender or significant interaction effects between the test condition and gender.

During the training process, we employed a three-down/one-up staircase method, which allowed the participants’ accuracy to converge at 79.4%. Additionally, we observed that the accuracy of the participants across all four angles during the post-test was above 90%. To further analyze the accuracy across the four angles in the post-test, a one-way analysis of variance (ANOVA) was conducted. The results revealed no significant differences in accuracy among the different angles (*F* (3, 84) = 2.64, *p* = 0.06). Thus, our training approach enabled the participants to maintain a high level of accuracy across all angles in the post-test. Therefore, we posit that the variation in participants’ improvement in terms of angles primarily manifests in their response times. To assess the degree of improvement in response times, a one-way analysis of variance (ANOVA) was conducted. The results revealed significant differences in the improvement of response times among the different angles (*F* (3, 84) = 5.05, *p* = 0.003). Post hoc Bonferroni multiple comparisons indicated that the improvement in response time at 0° angle was significantly lower than that at the other three angles, while there were no significant differences in the improvement of response times among the remaining three angles (see Figure 6).

We also conducted a two (test) × two (group) × two (gender) × four (angle) repeated measures analysis of variance (ANOVA) on reaction times to further investigate the specific effects of our training intervention on mental rotation ability within the experimental group. The results revealed a significant three-way interaction of test × angle × group (*F* (3, 38) = 8.99, *p* < 0.001, η_p_^2^ = 0.18). Further analyses showed that prior to training, there were no significant differences in reaction times between the control and experimental groups at any of the four angles. However, after training, there was no significant difference in reaction times at 0° between the control and experimental groups. In contrast, at the other three angles, the experimental group exhibited significantly faster reaction times compared to the control group (60°: *p* = 0.006; 120°: *p* = 0.003; 180°: *p* = 0.028). Additionally, no other four-way or three-way interactions were found to be significant.

## 5. Discussion

The present study employed computerized adaptive training to investigate the behavioral performance of male and female participants in a mental rotation task at different rotation angles, before and after training. Our analysis indicated the following: (1) the response time and accuracy of participants were integrated into the mental rotation task. We observed no significant differences between the control and experimental groups before training; however, after training, the experimental group showed significantly greater improvement compared to the control group. These results confirm the effectiveness of the training method proposed in this study for enhancing participants’ performance in mental rotation tasks; (2) before the training, the mental rotation test results revealed notable gender differences, with female participants exhibiting slower reaction times compared to male participants. However, following the training, these differences significantly reduced and became nonsignificant. It is important to note that this change in gender differences did not have a corresponding impact on the accuracy of responses; and (3) the effects of training at different rotation angles were examined, and the results demonstrated a significant improvement in participants’ performance at all four rotation angles, with the magnitude of improvement for the three angles involved in the training being significantly greater than that for the 0° angle. Furthermore, no significant gender differences were found in the training effects across different angles.

The behavioral results indicate that there were no significant differences in accuracy and response time between the experimental and control groups before training in the mental rotation task. However, after training, the experimental group exhibited greater improvements compared to the control group. These findings confirm the effectiveness of computerized adaptive training in enhancing mental rotation ability. This supports our hypothesis that the computerized adaptive training used in this study effectively improves mental rotation ability [35,36,37]. It is worth noting that the control group also showed improvements in mental rotation ability during the post-test. We speculated two reasons accounted for this phenomenon: firstly, the practice effect from the pretest. Although the control group did not receive training during the intermediate five days, they still gained some learning regarding stimulus materials and task rules through the identical tasks in the pretest. Consequently, the participants showed improved performance in the post-test due to the practice effect. Secondly, the regression to the mean effect. It is possible that the initial performance of the control group in the pretest was relatively lower than their actual ability level. As such, it is possible to observe the performance either improve or return to the average in subsequent evaluations.

Additionally, in line with previous research, we observed that prior to training, female participants performed inferiorly to male participants in the mental rotation task, but this gender difference in reaction time became nonsignificant after training. These findings align with the view that training has a more pronounced impact on enhancing mental rotation performance in females and can mitigate gender differences in this ability [27,28,29,31,38]. In order to make a more thorough investigation into the gender differences following the training and to eliminate the potential influence of a ceiling effect, a further examination was conducted [33]. We compared the reaction times of male and female participants for the four angles in the post-test and found no gender differences across all four angles. This indicates that even at the more challenging angles, the performance of female participants after training did not significantly differ from that of male participants. These findings suggest that training indeed attenuated the gender differences in mental rotation abilities, and the reduction in these differences is not solely attributable to ceiling effects in certain angles. We believe that the observed reduction in gender differences after training may be attributed to the potential benefits that females derive from the feedback on response times during the training [39]. Traditional stereotypes have suggested a male advantage in mental rotation abilities [40]. The provision of feedback during the training phase may encourage females to respond more quickly than in situations without feedback. Women seem to derive benefits from receiving feedback, which enhances their sense of security and self-confidence, leading to faster responses. Prior research has indicated that individuals with initially lower confidence levels often exhibit more significant improvements compared to those with higher confidence levels [41]. Therefore, in the future, we may consider conducting further investigations to explore the influence of participants’ confidence levels on the effectiveness of mental rotation training.

In the results, we focused primarily on differences in reaction time rather than accuracy. Several reasons account for this choice. Firstly, the adaptive training approach tailored the stimulus presentation to individual trait levels [42]. The mental rotation task employed in this study utilized a three-down/one-up staircase procedure to adaptively adjust the presentation time of stimuli, allowing participants to maintain a high and stable level of accuracy after five days of training. It is our contention that under this training paradigm, the participants’ goal is to achieve high accuracy while completing the training. During the post-test assessments, participants tended to sacrifice some reaction time in favor of higher accuracy when making speed–accuracy trade-offs. Indeed, we observed that participants achieved an accuracy rate of over 90% for all four angles in the post-test, indicating that our training protocol facilitated consistently high accuracy across different angles. Secondly, we believe that reaction time is a more indicative measure of mental rotation ability. Previous research has also highlighted that increased reaction time reflects the rotation process itself and, consequently, reflects the speed of mental rotation [43,44,45]. Reaction time represents the speed at which individuals process and mentally manipulate spatial information during mental rotation tasks. Faster reaction times suggest quicker cognitive processing and more efficient mental rotation abilities. On the other hand, accuracy primarily reflects the correctness of responses but may not capture the underlying cognitive processes or the speed of mental rotation. Mental rotation tasks often require individuals to mentally rotate objects or spatial stimuli to match a target orientation. The cognitive processes involved in mentally manipulating and transforming spatial representations play a crucial role in completing these tasks. Reaction time captures the time taken to execute these cognitive processes, providing insights into the efficiency and speed of mental rotation. Additionally, the relationship between reaction time and accuracy can be influenced by various factors, such as task difficulty, individual differences in cognitive abilities, and strategies employed by participants. Participants may prioritize accuracy over speed or vice versa, leading to trade-offs between the two measures. However, reaction time tends to be more sensitive to subtle differences in mental rotation ability, as even small variations in cognitive processing speed can result in noticeable differences in reaction time performance.

After training, participants exhibited more pronounced improvements primarily at the 60°, 120°, and 180° rotation angles, for which we propose two explanations. Firstly, in this study, we used 0° as the baseline and did not include a corresponding test task for the 0° rotation angle during the training phase. Participants showed significant improvements primarily at larger angles rather than at 0°, further highlighting the effectiveness of mental rotation training. If improvements were observed at the 0° angle, it would indicate that the training could enhance participants’ overall response speed, but it might not necessarily lead to a corresponding improvement in their mental rotation ability. However, our research results demonstrate that the training intervention effectively improved participants’ mental rotation abilities at these specific angles. Secondly, the 0° task was relatively easier due to its low difficulty level, which limited the room for improvement [15,46]. This task involved judging the similarity between left and right images, and as no rotation was required to determine the identical nature of the two images at 0°, the task was relatively straightforward, with minimal potential for improvement. Participants may have reached a performance ceiling for this task after training, thus limiting the ability to reflect the impact of the training from this angle. Future research could enhance the task difficulty by using a more diverse set of stimuli to explore whether the effects of training can be generalized to untrained angles. The improvement levels across the three angles involved in the mental rotation training are similar. We propose that this similarity may be attributed to two factors: transfer effects and the enhancement of comprehensive cognitive abilities. Firstly, through training, participants may develop more flexible and efficient mental rotation strategies that can transfer to different angles. The learning and application of these strategies can result in progress across various angles in tasks, indicating transfer effects. Secondly, the improvement of mental rotation ability is related to the improvement of comprehensive cognitive ability [47,48]. Training leads to improvements in attention, spatial perception, working memory, and other cognitive processes that play an important role in mental rotation tasks [49,50,51]. Therefore, the enhancement of comprehensive cognitive abilities may contribute to consistent progress across different angles in tasks. In conclusion, the effectiveness of mental rotation training may manifest consistently across different angles, which could be attributed to the combined effects of transfer effects and the enhancement of comprehensive cognitive abilities. However, further research and empirical evidence are needed to confirm the specific reasons.

The findings of this study hold implications for future research. Firstly, we employed a three-down/one-up staircase procedure to adjust stimulus levels based on participants’ responses. This training method helps overcome the limitations observed in previous studies that used fixed stimulus levels. Our results also indicate the practical utility of this training paradigm in promoting mental rotation abilities. Secondly, we explored the effects of training on participants’ performance in the mental rotation task from different angles and discussed the findings based on gender. We found that the gender difference in mental rotation ability diminished in terms of reaction times after training. However, this change in gender differences was not reflected in the accuracy of responses. It is possible that the gender difference in mental rotation ability did not manifest in accuracy measures. A more plausible explanation is that due to our training program, participants maintained higher accuracy rates in the post-test, which resulted in no significant change in gender differences in accuracy. This further supports our focus on changes related to reaction times. Regarding the angles, we found greater improvements for the three angles involved in the training compared to the untrained angle. It would be valuable for future research to further investigate whether training with more challenging rotation angles can be transferred to other angles. Thirdly, the population we trained in this study consisted of healthy young individuals. Existing research has primarily focused on spatial ability in patients and the plasticity of mental rotation in children, with limited studies on mental rotation training in healthy young adults. Additionally, by including participants from the normal population, our results can be generalized to a broader range of individuals, enhancing the external validity of the study.

The current study had several limitations. Firstly, it must be acknowledged that the control group in our study did not receive different training, which may limit a more rigorous comparison of the effectiveness of training regiments. Future studies could explore additional control conditions to reinforce the validity of the findings. Secondly, the pretest and post-test used the same mental rotation test material, which may have affected the participants’ familiarity with the test item. Although we set up a control group to eliminate this effect as much as possible, we can still use different picture materials to provide stronger evidence of training effects in future studies. Lastly, the exclusive use of two-dimensional (2D) materials for testing and training may restrict the applicability of the findings to real-world scenarios involving three-dimensional perception. Future investigations could incorporate three-dimensional (3D) stimuli or virtual reality environments to provide a more realistic assessment of mental rotation abilities. In conclusion, these limitations underscore the need for further research to advance our understanding of mental rotation abilities in a more comprehensive and rigorous manner, addressing the identified concerns and exploring alternative experimental designs.

The present study employed computerized adaptive training to investigate the behavioral performance of participants of different genders in a mental rotation task at different rotation angles, before and after training. The results indicate that training had a facilitative effect on the enhancement of mental rotation abilities for both male and female participants. In addition, the training was found to eliminate gender differences primarily in mental rotation reaction times, rather than accuracy rates. These findings are consistent with our hypotheses and demonstrate the effectiveness of the computerized adaptive approach used in this study for training mental rotation abilities. Regarding the angles, we observed that the magnitude of improvement was significantly higher for the angles involved in the training compared to the untrained angle. However, no significant differences in improvement were found among the three trained angles. It is important to note that there are limitations in the angle settings and the richness of the testing materials, which require further detailed investigation.

## Figures and Tables

**Figure 1 behavsci-13-00719-f001:**
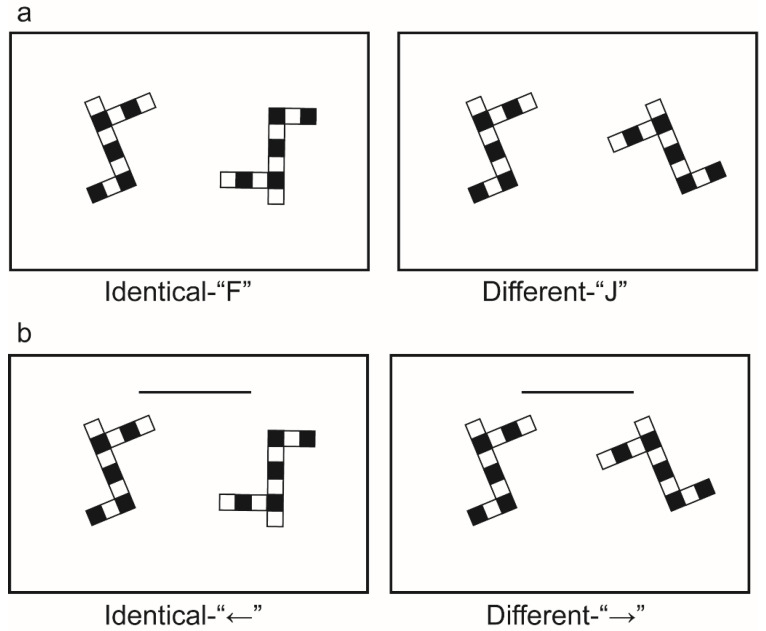
Examples of experimental stimulus materials: (**a**) mental rotation ability test task; (**b**) mental rotation training task. The experimental material consisted of black and white squares. A set of graphs consists of two two-dimensional graphs, one consisting of five black squares and six white squares. The horizontal dark line in (**b**) indicated the time progress bar. Participants need to react before the progress bar shrinks until it disappears. During the testing phase, we utilized a fixed-stimulus method, where the presentation time for each trial was consistent, thus obviating the need for a time progress bar. In the training phase, an adaptive approach was employed, and three different rotation angles were intermixed. As a result, it was necessary to include a time progress bar to alert participants to the time constraints.

**Figure 2 behavsci-13-00719-f002:**
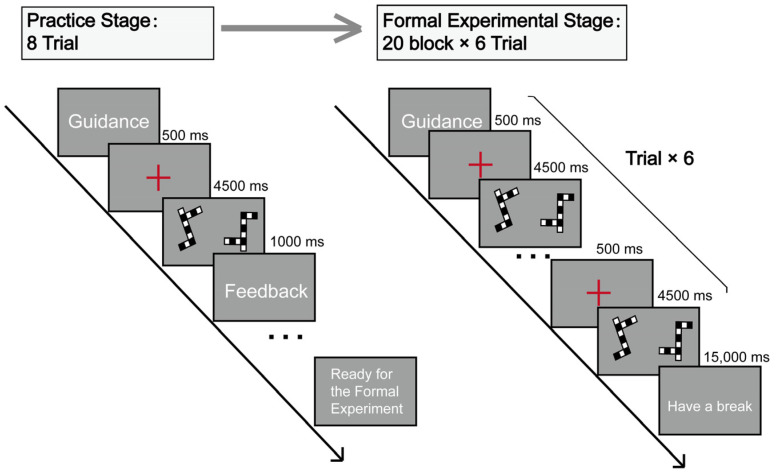
Flowchart of the experiment (mental rotation ability test task). A red “+” gaze point was presented before each trial. Each experiment presented a set of black and white squares with different relative rotation angles.

**Figure 3 behavsci-13-00719-f003:**
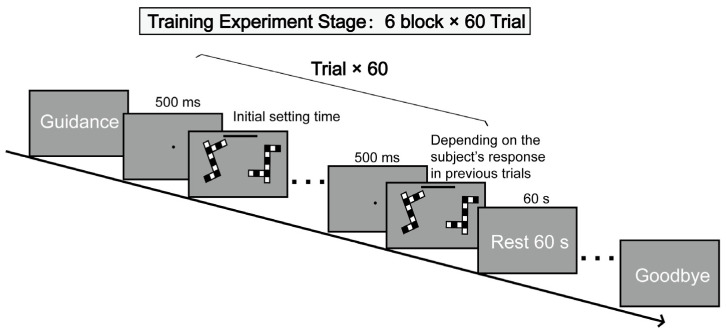
Flowchart of the experiment (mental rotation training task). Each experiment presented a set of black and white squares with different relative rotation angles. The initial setting time was 5 s for 60°, 7.5 s for 120°, and 10 s for 180°. A three-down/one-up staircase algorithm controlled the presentation time of the target image. This method decreased the presentation time by 10% if subjects correctly responded every three consecutive times and increased it by 10% if subjects made an incorrect response.

**Figure 4 behavsci-13-00719-f004:**
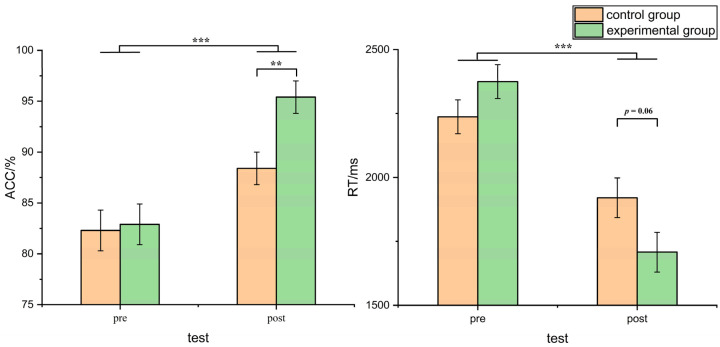
Comparison of reaction time and accuracy between control and experimental groups before and after training (error bars indicate standard errors; ** indicates *p* < 0.01; *** indicates *p* < 0.001).

**Figure 5 behavsci-13-00719-f005:**
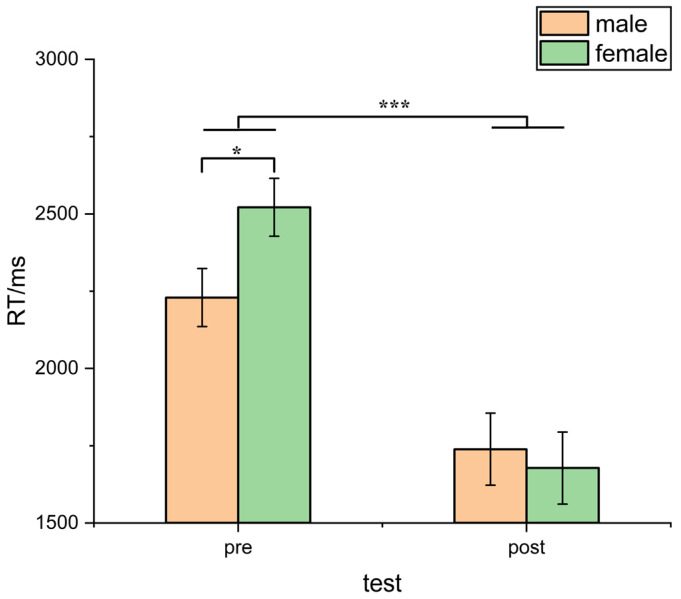
Comparison of reaction time between male and female participants before and after training (* indicates *p* < 0.05; *** indicates *p* < 0.001).

**Figure 6 behavsci-13-00719-f006:**
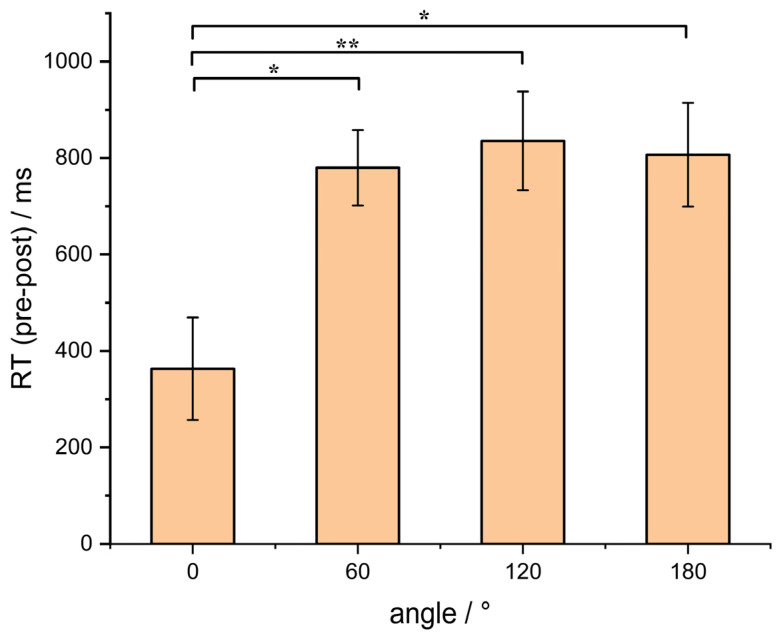
The improvement in reaction time at each angle (* indicates *p* < 0.05; ** indicates *p* < 0.01).

**Table 1 behavsci-13-00719-t001:** ANOVA results of response time improvement.

Factor Effects	RT		
	F	*p*	$\eta_{p}^{2}$
test	F (1, 20) = 82.874	<0.001	0.81
gender	F (1, 20) = 0.787	0.39	0.04
angle	F (3, 60) = 90.249	<0.001	0.82
test × gender	F (1, 20) = 5.432	0.03	0.21
test × angle	F (3, 60) = 13.608	<0.001	0.41
angle × gender	F (1, 20) = 0.259	0.85	0.01
test × gender × angle	F (3, 60) = 1.299	0.28	0.06

## Data Availability

The data that support the findings of this study are available from the corresponding author upon request.

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
