# Peer review of "Gender Differences in Mental Rotational Training Based on Computer Adaptive Tests"

_behavsci, 2023, doi:10.3390/bs13090719_

Round 1

Reviewer 1 Report

Dear authors,

Below are the  comment on your manuscript:

1. To clarify the research gap in greater detail

2.  To explain the fundamental of the study by Levitt H (1971) that was used as the source for this study. It is good to describe a brief three down /one up staircase method in the introduction section

3. State the implication of this study since this study was conducted among  normal population

4. Described  brief sampling and randomized method used in the study 

Reviewer 2 Report

Overall, this paper is a well-written and gives an important contribution to the specific research field. I only ask an extention of references to improve the quality and the strong of this work.

Moderate editing of English language required

Reviewer 3 Report

Review

Gender Differences in Mental Rotation Training based on Computer Adaptive Tests

In this study, students were trained with a mental rotation task to improve their mental rotation abilities. Compared to a control group, trained participants rotated the objects faster and gender differences in reaction time disappeared.

There are already many studies regarding the malleability of spatial abilities. However, this adaptive training seems promising. While I enjoyed reading the manuscript, there are some queries that have to be addressed before publishing.

First paragraph: I would add the study of Wai et al. (2009).

Line 61: I would add the meta-analysis of Voyer et al. (1995).

Line 47: What is meant by “the training paradigm lacks intelligence”?

Line 94: You write that you tested adolescents representing diverse gender groups. In line 111 it says that the participants were 44 volunteers and you only report the number of males. Are the rest females or were there diverse people? With a mean age of 21, I would not describe them as adolescents. Please also describe how many participants (gender, age) were in the training and in the control group.

Line 127: I would add that the objects were rotated in the picture plane (2D).

Line 153: did the feedback “wrong” only appear when participants made no answer in the given time? Or was there also feedback (right/wrong) on the answers?

Line 157: You write that there are 20 blocks and a pause was given after two blocks. What is specific in a block? Did they consist of the same objects? Why not write that there are ten blocks with a pause after each?

Line 173: Does the time progress bar illustrate the remaining time in the whole test or for each item?

Line 175: Participants could use the three-down/one-up staircase algorithm as feedback because the time changes if the answer was wrong. This could influence males and females differently (see Rahe et al., 2019). 

Figure 1: What was the initial setting time? Was it 4500ms?

Line 172 and line 185: you write that there was a total of 360 sets with three angles and that each angle consisted of 80 trials. Why aren’t that 240 trials?

Results, 4.1: I suggest calculating a repeated measure ANOVA instead of a t-test with differences and illustrating the results with bars for the control and the training group. Additionally, please report effect sizes.

Table 1: The effect sizes are very large. Please check that these are really partial eta-squares. 

Results 4.2: Why do you analyze the improvements only for the experimental group and not compared to the control group?

I would drop Table 2. It does not give any more information than you already provide in the text.

Line 256: you found this result (2) only for reaction time but not for accuracy. Please add that to the discussion (also in line 358, 383)

Lines 323-333 (reasons why participants mainly improved at larger angles but not at 0°): I would add that this illustrates the effectiveness of your training in mental rotation. If the angle of 0° improved, your program would improve participants’ overall response speed but not their mental rotation ability.

Limitations:

The control group was not trained in different training. 

The same mental rotation test was used as pre- and post-test.

References:

In some references, you mixed up first and surnames (1,4,27).

Suggested literature:

Rahe, M., Ruthsatz, V., Schürmann, L., & Quaiser-Pohl, C. (2019). The effects of feedback on the gender differences in the performance in a chronometric mental-rotation test. Journal of Cognitive Psychology31(4), 467-475.

Voyer, D., Voyer, S., & Bryden, M. P. (1995). Magnitude of sex differences in spatial abilities: a meta-analysis and consideration of critical variables. Psychological bulletin117(2), 250.

Wai, J., Lubinski, D., & Benbow, C. P. (2009). Spatial ability for STEM domains: Aligning over 50 years of cumulative psychological knowledge solidifies its importance. Journal of educational Psychology101(4), 817.

Round 2

Reviewer 3 Report

The authors have answered all of my comments.

The manuscript has improved a lot and is now suitable for publication.